# Is It Useful to Repeat Blood Cultures in Endocarditis Patients? A Critical Appraisal

**DOI:** 10.3390/diagnostics14141578

**Published:** 2024-07-22

**Authors:** Wouter Kok

**Affiliations:** Department of Clinical and Experimental Cardiology, Amsterdam Cardiovascular Sciences, Meibergdreef 9, 1105 AZ Amsterdam, The Netherlands; w.e.kok@amsterdamumc.nl

**Keywords:** endocarditis, blood cultures, bacteremia, persistent infection

## Abstract

Background: Previous guidelines for endocarditis have suggested repeating blood cultures until they become negative, with limited evidence. Methods: Literature reviews were conducted (1) on the incidence of persistent bacteremia and association with outcome and (2) on timing of valve culture negativization to examine the claim for prolongation of antibiotic therapy starting from negative blood cultures. Results: Persistent bacteremia and fever may be present in the first 3 days of endocarditis, despite treatment, and are more common in Staphylococcus (especially MRSA) and Enterococcus species. Persistent bacteremia (48–72 h), persistent infection (day 7), and new onset septic shock are related and predict in-hospital mortality. It is, however, persistent infection at day 7 and septic shock that primarily determine the infectious course of endocarditis, and not persistent bacteremia. Valve cultures at surgery become negative in most cases (>85–90%) after 14–21 days of antibiotic therapy, with no calculated benefit for prolonging therapy after 21 days. Conclusions: Persistent infection at 7 days after appropriate antibiotic therapy is a better key event for prognosis then positive or negative blood cultures at 48–72 h. Therapy prolongation from the day of negative blood cultures is not reasonable. There is no need to survey blood cultures in endocarditis patients after starting therapy.

## 1. Introduction

The American Heart Association and European Society of Cardiology (ESC) guidelines on endocarditis from 2015 have introduced a recommendation to repeat blood cultures every 1–2 days after the initial diagnosis of endocarditis until they become negative during antibiotic therapy [1,2]. The recommendation received a class IIa indication in the American guideline of 2015 [1] and was a key message in the ESC guideline of 2023 [3]. The most simple answer to the question why blood cultures should be repeated in endocarditis patients is that a negative follow-up culture confirms the effectiveness of the therapy [2]. An added rationale is present in both guidelines that therapy duration may be calculated from the day that the blood culture becomes negative [1,2,3]. However, the question at which time its counterpart, persisting positive blood cultures, is sufficiently pointing towards a diagnosis of persistent infection (a reason for further investigation) or uncontrolled infection (an operation indication) has not been answered definitively. Second, the rationale of determining therapy duration from the day of negative culture is provided without evidence, level C, in both guidelines. Therefore, routinely repeating blood cultures after starting antibiotic therapy in all patients with endocarditis to seek for a negative blood culture may become a costly affair without any proven consequence. To shed some light on possible misunderstandings, the present review was written. A short population, intervention, comparison, outcome (PICO) Pubmed search was performed on the relation between persistent bacteremia and outcomes in endocarditis with search terms “humans”, “female”, “male”, “endocarditis” (population), “bacterial culture”, “bacteremia”, “blood culture”, “drug monitoring” (as intervention terms), “case-control studies”, “cohort studies”, “retrospective studies”, “prospective studies”, “multivariate analysis” (as comparison terms) and “bacteremia/mortality”, “Endocarditis, Bacterial/mortality”, “prognosis”, “survival analysis”, “treatment outcome” (as outcome measures). A second PICO on timing of negativization of valve cultures in patients who underwent surgery for endocarditis was performed with search terms “humans”, “female”, “male”, “endocarditis” (population), “endocarditis, bacterial/blood”, “endocarditis, bacterial/microbiology”, “endocarditis, bacterial/drug therapy”, “cardiac surgical procedures/methods”, “cardiac surgical procedures/standards*” (intervention), “case-control studies”, “cohort studies”, “retrospective studies”, “prospective studies” (as comparison terms), and “heart valves/microbiology*”, “treatment outcome” (outcome measures). For full search queries, see the Appendix A. In addition, American and ESC guidelines on endocarditis were reviewed, especially those of 2015 [1,2], and the ESC guideline on endocarditis 2023 [3]. The present review was performed by a single reviewer and, despite efforts to be comprehensive, was not a systematic review.

In the present paper, the findings of persistent bacteremia pertain only to patients diagnosed and treated as endocarditis, because this is the group of patients addressed in the guidelines mentioned. The question of repeating cultures has also been discussed in reports on bacteremia of unknown cause, where persistent bacteremia may raise a suspicion of a diagnosis of endocarditis [4,5]. In patients with a diagnosis of possible, probable, or confirmed endocarditis, the therapy for endocarditis has already been started and the question is whether persistently positive blood cultures would be lead to a useful change in therapy or approach. For practical purposes and also for future studies, it may help to interpret persistently positive blood cultures within the definitions of persistent bacteremia and try to relate them to persistent infection and uncontrolled infection as defined in the guidelines.

## 2. The Definitions of Persistent Bacteremia

In 1994, the Duke criteria were published for the diagnosis of bacterial endocarditis, and contained a major criterium of “persistent bacteremia” [6]. This could be either three repeated positive blood cultures with at least one hour separated between the first and last culture, or two repeated positive blood cultures when the second was performed >12 h after the first blood culture [6]. The usual duration of blood cultures at this time may not have been <24 h after admission, and antibiotics could have been started when the diagnosis of endocarditis was highly suspected [7].

In 2005, a large international study described that the abovementioned definition of persistent bacteremia >12 h after the first blood culture was found in 7.5% of patients and that it was particularly associated with the presence of a methicillin-resistant *Staphylococcus aureus* (MRSA) which on its own had a risk of 42.6% of persistent bacteremia [8]. Other associations with persistent bacteremia in their study were healthcare-associated endocarditis (indwelling catheters and hemodialysis shunts) and the use of vancomycin, which had a known delayed activity against Staphylococcus aureus [8]. Finally, persistent bacteremia in 17% of the subset of 300 *Staphylococcus aureus* endocarditis patients was independently associated with a threefold increased risk of in-hospital mortality [8]. In another substudy, persistent bacteremia (same definition as above) in a subset of 556 patients with prosthetic valve endocarditis was found in 8.8% and was associated with an adjusted fourfold increased risk of in-hospital mortality [9]. Despite the limitation of the inexact time period (>12 h after the initial blood culture) in the definition on persistent bacteremia, these two studies confirm the prognostic importance of persistent bacteremia.

A more clearly delineated definition of persistent bacteremia was given in a retrospective study on 407 patients with left-sided endocarditis, in which persistent positive blood cultures at 48–72 h after the initiation of antibiotic therapy were demonstrated in 89 (35%) of 256 patients with initial positive blood cultures [10]. *Enterococcus*, coagulase-negative staphylococci, and *S. aureus* species were the dominant group of bacteria in those with persistent positive blood cultures [10]. Persistent blood cultures were associated with prosthetic valve endocarditis, and with somewhat higher rates of periannular complications and atrioventricular block [10]. Finally, in multivariate analysis, persistent positive blood cultures at 48–72 h had a twofold increased risk of in-hospital mortality (44% versus 25%), independent from the presence of staphylococcus aureus etiology, heart failure, prosthetic valve endocarditis, or periannular complications [10]. The outcome of patients undergoing surgery (60% of those with persistent positive blood cultures) was worse when patients had persistent positive blood cultures than after negativization of blood cultures (in-hospital mortality 41% versus 27%). However, those with persistent positive blood cultures undergoing surgery had similar in-hospital mortality to those not undergoing surgery (in-hospital mortality 41% versus 47%), making these retrospective comparisons more difficult to understand [10]. In the American and ESC guidelines a class IIa recommendation for urgent surgery is given for persistent positive blood cultures after 7 days, when not explained by other sources. In the ESC guideline 2015, it was suggested that surgery should be considered for persistent positive blood cultures after 72 h, when similarly not explained by other sources (see Table 1) [2,3].

## 3. Definitions and Consequences of Persistent and Uncontrolled Infection

Apart from persistent bacteremia, there is also persistent fever as a sign of ongoing infection, and fever for 7 days or longer during endocarditis treatment has been noted as a sign of a complicating cardiac infection [11,12]. It is probably from this experience that Gutschik et al. stated in 1998 for one of the first guidelines for endocarditis that “unchanged febrile illness beyond the first week of treatment” was a warning sign serious enough to “warrant a reassessment of the antibiotic prescription”, and “draw special attention to a possible complicated endocarditis with valve ring abscess” [7].

In both ESC and American guidelines of 2015, persistent infection is defined, but the definitions are not completely similar. In the ESC guideline, it is defined as fever *and* positive blood cultures >7–10 days after starting antibiotic therapy [2,3]; in the American guideline, this is fever *or* persistent blood cultures >5–7 days after starting antibiotic therapy [1] (see Table 1). In both guidelines, persistent infection first of all has the consequence that further investigation is needed for an extracardiac source (septic embolization or line infection) before deciding on urgent surgery [1,2]. In the ESC guideline 2015, the comment was added that the period of 7–10 days in persistent infection was arbitrary and that more than 3 days with persistent positive blood cultures would already be a reason for additional investigation of its causes [2,3]. This comment was not added in the American guidelines of 2015, possibly because of an already shorter period of 5–7 days for persistent infection as an incentive for further investigation [1]. It should also be noted that persistent infection not only means positive blood cultures *and* persistent fever but that it can also mean persistent fever in patients with negative blood cultures; hence, the term “*or*” seems more appropriate in the American guideline. Negative blood cultures may not always indicate source control in the sense of negative valve tissue cultures, so other signs of persistent infection should still be looked for despite negative blood cultures.

Uncontrolled infection is defined in the ESC guideline 2015 and 2023 as either persistent infection, not responding to required interventions, or as perivalvular extensions such as pseudoaneurysms and abscess, or as infections resistant to antibiotics such as fungal infection, and is a reason for urgent surgery [2,3]. In the American guidelines, the term uncontrolled infection is not used; instead, the conditions leading to urgent surgery are all mentioned separately [1].

It can be difficult in retrospective studies to determine how to interpret uncontrolled infection as the primary reason for surgery, as there are subgroups of uncontrolled infection. One of the subgroups is persistent infection (fever or persistent blood cultures) which, however, may coexist as surgical indications for operations, for example, in 17% of patients with heart failure [13] and in 28% of patients with periannular complications [14]. Using the definition of uncontrolled infection of the ESC guidelines 2015, three retrospective studies have attempted to estimate the impact of not performing surgery in those patients with uncontrolled infection as the indication for surgery [15,16,17]. In the study by Garcia Grania et al. on 405 left-sided endocarditis patients who underwent surgery, uncontrolled infection was the primary reason for operation in 31% and a secondary reason (next to heart failure) in 26%; heart failure as the only reason for surgery was present in 28%, and prevention of embolism was the reason for surgery in 14% [16]. Patients with uncontrolled infection were, however, less likely to undergo surgery than patients with heart failure (54% versus 74%), which is largely explained by high prevalences of septic shock and renal failure in those who are not selected for surgery [16]. Similar findings were reported by Ramos-Martinez [15]. The studies argue for surgery but encounter difficulties in comparing patients with and without barriers for surgery. All three studies point to a gap in the evidence on how to identify patients with uncontrolled infection at an early stage when surgery is still safe. The evidence gap will also and in particular concern those patients with septic shock at presentation and those with persistent infection after the first week in whom there is no other treatable cause [15,16].

## 4. What Is the Role of Repeating Blood Cultures to Identify Those at Risk for Persistent Infection?

In the pivotal study of Lopez et al. (2013), in 256 endocarditis patients in whom blood cultures were repeated at 48–72 h, persistent infection at 7 days was present in 49% of those with persistent positive blood cultures. It was also present in 32% of those with negativization of blood cultures [10]. The ESC guidelines cite this study as the reason to act on these persistently positive blood cultures because they are also associated with a twofold risk of in-hospital death [2,3]. The first question is, however, whether a 48–72 h blood culture is a representative culture defining the “normal” from the “abnormal” condition, and which are the variables that affect the persistence of bacteremia.

Persisting bacteremia can be seen to depend, first of all, on the type of bacteria and on the antibiotic given. For *S. aureus*, the question of persistent bacteremia was first answered in a randomized therapeutic study in 1982 for *S. aureus* endocarditis in 30 persons described as nonaddicts; cultures persisted for 2.8 ± 1.4 days for nafcillin plus gentamycin, and 4.1 ± 1.4 days for nafcillin alone [18]. The positive culture time corresponded with the time to become afebrile. A second study was performed in 1991 in 40 patients with MRSA endocarditis, randomly comparing therapy by vancomycin versus vancomycin plus rifampicin [19]. The median duration of bacteremia was 9 days (7 days for group I and 9 days for group II). The median duration of fever for all patients and for each treatment group was 7 days [19]. They stated that “although patients had sustained bacteremia, no unusual complications were seen in either treatment group, and most patients responded to continued antibiotic therapy” [19]. Other studies in *S. aureus* endocarditis confirm that cultures remain positive despite appropriate therapy for a median of 2–5 days [20,21,22]. Two of these studies report the 3-day cultures to be positive in 27% and 22% of endocarditis patients respectively, mainly those with MRSA endocarditis [20,21]. In those with MRSA endocarditis, persistent bacteremia for 3 days or more was present in 65%, accompanied by fever for an average of 6.6 days [20]. In those with methicillin-sensitive *S. aureus*, persistent bacteremia was present for 3 or more days in 9% of patients, accompanied by fever for an average of 4.7 days [20]. Two other larger studies on MRSA endocarditis, treated with vancomycin, report persistent bacteremia after 3 days in 46% of patients [23] and persistent bacteremia after 7 days in 59% [24]. The importance of antibiotic resistance and the minimal inhibitory concentration (MIC) value of the antibiotic vancomycin is particularly relevant for MRSA endocarditis [23].

*Enterococcus faecalis* is prevalent among the causative agents of endocarditis but is also known for partial antibiotic resistance against penicillin and ampicillin, and recently emerging high-level resistance to aminoglycosides [25]. In *Enterococcus faecalis,* endocarditis persistent bacteremia >48 h after antibiotic therapy was present in 28% in a small study of 7 patients [26]. In another study of 113 *Enterococcal* endocarditis patients, persistent bacteremia at 48–72 h was present in 24.8% [27]. Only one study from the Mayo clinic in 85 *Enterococcus faecalis* endocarditis patients showed no persistent bacteremia after 48 h after treatment with ampicillin combined with penicillin or combined with gentamycin [28]. In a fourth study of 109 patients with native valve *Enterococcus faecalis* endocarditis, treated with ampicillin and ceftriaxon, persistent bacteremia >7 days was found in 16 patients or 14.7%; persistent fever was not reported [29].

For other bacteria, such as *Streptococcus bovis* and *Streptococcus viridans*, persistent bacteremia was seen at 48–72 h in 17% and 12.7%, respectively [27]. In a recent large study of 159 non-staphylococcal endocarditis patients, persistent bacteremia >48 h was encountered in only 7 patients (4.4%), of whom 5 had *Enterococcus faecalis* endocarditis [30]. In those with *Streptococcal* endocarditis, no persistent bacteremia was found. In their study, fever persisted for a median of 2 days (range 0–5) [30].

In summary, with regard to the persistence of bacteremia *and* fever following the start of antibiotic therapy in endocarditis, older and new reports [18,19,20,30] underline that bacteremia and fever may persist for 2–5 days in certain causative agents such as *Staphylococcus aureus*, for 3–9 days particularly in MRSA, and between 2 and 7 days for certain *Enterococcal* species. Also, prosthetic valve endocarditis had a slightly higher incidence of persistent bacteremia at 48–72 h than native valves (43% versus 29%) [10]. There is, therefore, a rather variable time of persistent bacteremia that depends on the bacterial result of the initial blood culture and the antibiotics given, and possibly also on the presence of native versus prosthetic valves. This variability in culture results, perhaps in most cases representing “normal” findings, would not support an early time of 3 days of positive blood culture persistence in all patients as the time to attempt to define persistent bacteremia as persistent infection. Early investigations of persistent infection (among which to confirm the presence of the initial culture in repeated blood cultures) may, however, still be useful in those with bacterial species that are difficult to treat, or can be useful as surrogate outcomes for studies comparing antibiotic regimens.

## 5. The Incidence, Risks and Associations of Persistent Infection

There is still the key finding of persistent infection and in-hospital mortality, predicted by positive blood cultures at day 48–72 h [10], that needs more explanation. From a Spanish three-center cohort of 894 episodes of endocarditis (between 1996 and 2011), the incidence and consequences of persistent infection were reported systematically [10,13,31]. In all three studies, the definition for persistent infection was persistent fever *and*/*or* positive blood cultures after 7 days of appropriate antibiotic treatment, and after having ruled out other possible foci of infection. In the substudy of 256 endocarditis patients in whom blood cultures were repeated at 48–72 h, persistent infection at 7 days was present in 37.5%; persistent infection at 7 days was present in 49% of those with persistent positive blood cultures [10]. Persistent positive blood cultures carried a twofold risk for in-hospital mortality. A peculiarity of the analysis on prediction of in-hospital mortality that is worth mentioning here is that the prediction analysis was conducted on the full population of 407 patients, but that follow-up cultures were only performed in 256 patients, introducing selection bias [10]. The results certainly remain interesting, however, and the study by itself raised the urgency bar for timely treating endocarditis. In a parallel study on septic shock as outcome, persistent infection after 7 days was seen in 33.1% of 842 episodes of endocarditis in whom there was no septic shock at presentation [31]. In the third study on 89 endocarditis patients undergoing urgent surgery, persistent infection was found in 28 (31%), with persistent infection as primary reason to operate in 17 patients (19%) [13]. Most of the other patients underwent urgent surgery because of heart failure [13].

The key element in prediction of outcomes, therefore, is proposed to be either persistent bacteremia at 48–72 h or persistent infection at day 7, but it is difficult to compare the study results, because the first study of Lopez et al. [10] used only the key element of positive blood cultures at 48–72 h as a risk predictor while mentioning persistent infection (see central graphical illustration). The second study of Olmos et al. [31] used the key elements of persistent infection and new onset septic shock, but not persistent bacteremia. However, they can be compared as to the selection of patients at risk, and the magnitude in which these predictions lead to separation of in-hospital mortality risk. The scenario of positive blood cultures at 48–72 h in one-third of patients, predicting persistent infection, has an outcome difference in mortality of 44% versus 25% (see central graphical illustration). The second study of Olmos et al. found (in their Supplemental Table 4) that persistent infection at day 7 in endocarditis patients carries the risk of new onset septic shock (28% vs. 4.4% in those without persistent infection) [31], with new onset septic shock being related to an increase in in-hospital mortality (80% versus 18%) [31]. From the data on (relatively) low mortality rates in those without new onset septic shock (the majority of patients), one would rather select the second scenario, if one argues for a more precise predictor of in-hospital mortality, while selecting the same one-third of patients at risk. The main clue here is that it is primarily septic shock that determines mortality from an infectious cause. Other predictors of new onset septic shock are diabetes mellitus, *S. aureus* infections, acute renal insufficiency, and vegetation size ≥15 mm [31]. Periannular complications are associated with in-hospital death [10,14,31] and have increased rates of 48–72 h bacteremia [10], but do not seem to be related to new onset septic shock [31] or persistent infection [14]. Thus, for periannular complications, persistent bacteremia only seems to be a risk indicator, but not because of the ongoing infectious state; it might be explained by association with bacteria that are more often involved as causes of periannular complications [14].

In conclusion, it appears that persistent infection at day 7 more clearly defines risk of in-hospital mortality by the intermediate outcome of septic shock (that one would like to prevent) than positive blood cultures at 48–72 h that—although they are associated with persistent infection—have lower predictive value (central graphical illustration). It may be that a positive blood culture at 48–72 h may be interpreted too soon as being a sign of persistent infection, while those with negative blood cultures may be interpreted falsely as having cleared the risk of persistent infection. Evidently, there is still a gap in knowledge in early risk assessments of those who will be candidates for urgent surgery for the reason of persistent infection. Antibiotic regimens may have to be improved for more difficult-to-treat bacteria.

## 6. Why Do We Need Negative Blood Cultures as the Starting Point of Antibiotic Therapy Duration?

“Negative blood cultures, on the other hand, do not support a claim of bacteriologic cure of infective endocarditis, and thus no monitoring of efficacy is recommended” [7]. In the original endocarditis guideline of 1998, advice was given against routinely repeating blood cultures after initiating antibiotics for endocarditis [7]. This advice is in fact supported by the finding that persistent infection was still possible in 32% after negativization of blood cultures performed at 48–72 h [10]. Clearing the bloodstream from bacteria (with a first negative culture) as a starting point for determining the duration of antibiotic therapy, as indicated in some parts of the guidelines [1,2,3], is not evidence-based. It should be realized that a negativization of blood cultures does not represent a negativization of valve culture, and it may be worthwhile to know if prolongation of antibiotic therapy is even necessary from the perspective of the time needed for bacterial clearance of the affected valve cultures.

## 7. What Is the Time Needed to Clear Bacteria from Affected Valves in Endocarditis?

There are eight studies on valve culture in relation to the antibiotic therapy duration [32,33,34,35,36,37,38,39] (Table 2). All studies agree on the >85–90% likelihood of valve clearance after 14–21 days of appropriate antibiotic therapy.

In a study on valve cultures performed in 231 patients with endocarditis undergoing surgery, the time needed to have negative valve cultures in almost all patients was quite complete after about 14 days, with further prolonging antibiotic therapy beyond 21 days being of no added value [37]. The time to valve culture clearance was dependent on the bacterial species, with more time needed for *Enterococcus* and *Staphylococcus* species [37]. The results from logistic regression analysis were interpreted as that patients at higher risk of positive valve cultures were less likely to benefit from extended periods of antibiotic therapy beyond 3 weeks and that in case of persistent infection, this would probably make them earlier surgery candidates [37].

The main measure to prevent early nonresponse on the medication given is choosing the right antibiotic and dose, which is usually confirmed or adjusted after performing an in vitro test to assess susceptibility [7]. It appears that antibiotics given at the start of endocarditis are the effective ones in 75% of patients, and it should be remembered that the use of effective therapy is incorporated in the definitions of persistent bacteremia and persistent infection as it is defined, despite “appropriate antibiotic therapy” [30].

Notwithstanding the main limitation of the present critical review, that it is not a systematic review, the utmost effort was made to provide a rational and evidence-based review.

## 8. Conclusions

Persistent bacteremia after treatment for endocarditis is common during the first 3 days and is often accompanied by persistent fever, especially in certain bacterial species such as *S. aureus* and *Enterococcus* species. The key element for predicting septic shock and in-hospital mortality is persistent infection at 7 days, and not persistent bacteremia at 48–72 h. Antibiotic therapy prolongation from the day of negative blood cultures is not reasonable, with valve cultures at surgery being negative after 14–21 days in most cases, and therapy duration of 28 days as a minimum duration in most cases. In addition, in those cases in whom prolongation of antibiotic therapy would be considered because of persistent infection, a switch in antibiotic regimen or an indication for surgery is probably the better consideration. Standard surveying of blood cultures after starting antibiotic therapy in endocarditis patients should therefore not be advised.

## Figures and Tables

**Table 1 diagnostics-14-01578-t001:** Definitions of persistent and uncontrolled infection in American and ESC guidelines 2015.

	AHA Guideline 2015	ESC Guideline 2015
** *Persistent infection* **	Persistent bacteremia or fever lasting >5–7 days despite appropriate antibiotic therapy.	Persisting fever and positive blood culture (>7–10 days) despite an appropriate antibiotic regimen.
Action required	Excluding other sites of infection and fever.	Replacement of i.v. lines, repeat laboratory measurements, blood cultures, echocardiography, and the search for an intracardiac or extracardiac focus of infection.
	Surgery is indicated when persistent bacteremia or fever lasting >5–7 days despite appropriate antibiotic therapy and provided that other sites of infection and fever have been excluded.	Surgery has been indicated when fever and positive blood cultures persist for several days (7–10 days) despite an appropriate antibiotic regimen and when extracardiac abscesses (splenic, vertebral, cerebral, or renal) and other causes of fever have been excluded. However, the best timing for surgery in this difficult situation is unclear. Recently it has been demonstrated that persistent blood cultures 48–72 h after initiation of antibiotics are an independent risk factor for hospital mortality (Lopez 2012 [10]). These results suggest that surgery should be considered when blood cultures remain positive after 3 days of antibiotic therapy, after the exclusion of other causes of persistent positive blood cultures (adapted antibiotic regimen).
** *Uncontrolled infection* **	AHA guidelines do not use the term uncontrolled infection and present individual reasons for surgery. Persisting fever and positive blood cultures (>5–7 days), provided that other sites of infection and fever have been excluded, is one of them.	Locally uncontrolled infection (increasing vegetation size, abscess formation, false aneurysms, and the creation of fistulae) OR persisting fever and positive blood culture (>7–10 days), OR infection due to fungi or multiresistant organisms or in the rare infections caused by Gram-negative bacteria.
Action required	Surgery is recommended for uncontrolled infection.	Surgery is recommended as soon as possible. Rarely when there are no other reasons for surgery and fever is easily controlled with antibiotics, small abscesses or false aneurysms can be treated conservatively under close clinical and echocardiographic follow-up.

**Table 2 diagnostics-14-01578-t002:** Studies on the association of antibiotic duration and positive valve cultures at surgery.

Study	Number of Patients and % Native Valves	Number of Patients with Positive Valve Culture (% of Total)	Influence of Duration of Antibiotic Treatment on % Positive Valve Culture
Morris 2003 [32]	480 (62% native)	130 (30%)	In terms of standard duration of antibiotic treatment completed: ≤50%: 116/214 (54.2%) >50%: 14/145 (9.7%)
Upton 2005 [33]	131 (66% native), with *Streptococcal* endocarditis	25 (19%)	≤14 days: 24/69 (34.8%) >14 days: 1/62 (1.6%)
Mekontso Dessap 2009 [34]	90 (79% native)	46 (51%)	≤7 days: 35/45 (78%) >7 days: 11/45 (24%)
Voldstedlund 2012 [35]	223 (85% native)	58 (26%)	<14 days: 53/157 (33.8%) ≥14 days: 5/74 (6.7%)
Halavaara 2019 [36]	87 (87% native)	19 (22%)	<14 days: 19/53 (35.8%) ≥14 days: 0/34 (0%)
Gisler 2020 [37]	231 (66% native)	58 (25%)	≤7 days: 31/60 (52%) >7 days: 27/171 (15.8%) ≤15 days: 47/125 (37.6%) >15 days: 11/96 (11.4%) ≤21 days: 53/169 (31.4%) >21 days: 5/62 (8.1%) In logistic regression analysis, contribution of antibiotic duration per 2 days on prediction of positive valve culture is absent after 21 days. Other strong predictors for positive valve cultures are *Enterococcus* spp. and *Staphylococcus* spp.
Fillâtre 2020 [38]	148 (81% native)	46 (31%)	<14 days: 34/73 (46.6%) ≥14 days: 12/75 (16.0%)
Johansson 2023 [39]	345 (73% native)	78 (23%)	≤11 days: 73/208 (35.1%) >11 days: 5/137 (3.6%)

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
