# Peer review of "Is It Useful to Repeat Blood Cultures in Endocarditis Patients? A Critical Appraisal"

_diagnostics, 2024, doi:10.3390/diagnostics14141578_

Round 1
Reviewer 1 Report
Comments and Suggestions for Authors
Author Response
Reviewer 1
Thank you very much for the input and for careful reading.
- Line 107-110: In current guidelines (ESC 2023) a class IIa recommendation for urgent surgery is given for persistent positive BC after 7 days. Please rephrase it.
Authors answer: The specific sentence (other than above) was meant to capture that the study results of Lopez et al (ref 10) on persistent positive BC at 72 hours was only supported in the ESC guideline 2015 (see table 1). I rephrased the sentence because it may have led to confusion about what is and what is not recommended. The sentence is changed to: < In the American and ESC guidelines a class IIa recommendation for urgent surgery is given for persistent positive blood cultures after 7 days, when not explained by other sources. In the ESC guideline 2015, it was suggested that surgery should be considered for persistent positive blood cultures after 72 hours, when similarly not explained by other sources (see table 1).>
- A limitation section should be added, and conclusions should be partially rephrased according to these limitations. According to my opinion, the most important limitation is related to the absence of systematic methodology of literature searching. Systematic reviews with/without metanalysis could better answer those relevant clinical questions and this should be stated by the authors.
Authors answer: I agree with the reviewer that the current review is not systematic. I added a paragraph in introduction after line 57, explaining that the literature review itself was performed by a single reviewer: <The present review was performed by a single reviewer and, despite efforts to be comprehensive, was not systematic>, and just before conclusions that “Notwithstanding the main limitation of the present critical review, that it is not a systematic review, the utmost effort was done to provide a rational and evidence based review.”.
- Conclusions line 325-328: this sentence should be rephrased. Also positive BC at 48-73 h have shown to be independently associated to increased in-hospital mortality in IE as shown in the study by Lopez et al. (ref number 10 in the manuscript). Moreover, the sentence stating that positive BC at day 7 cannot be reliably predicted by positive BC after 48-72 h could be reconsidered. Nowadays there are no many studies comparing these two findings in their predictions of in-hospital mortality or persistent infection as also stated by the authors at line 253-255.
Authors answer: The sentence in lines 325-328 “persistent infection at 7 days which has been demonstrated to have the risk of new onset septic shock and hospital mortality, which cannot be reliably predicted by either persistent positive or negative blood cultures at 48-72 hours” should mean that 1) persistent infection at day 7 is the key element in predicting new onset septic shock and mortality, and 2) that a reliable prediction of persistent infection and mortality cannot be given by blood cultures at 48-72 hours. This latter point is made because too much focus was given in the study of Lopez et al on positive blood cultures, carrying a persistent infection risk of 49% and mortality risk of 44%, while negative blood cultures can still be followed by persistent infection in 32% and mortality in 25%. When you focus on persistent infection at 7 days, then the risk prediction (of sepsis and mortality 80% versus 18% in those without persistent infection) is much better, independent of previous blood cultures. These results come from the same study, so they are to be compared (only with smaller sample size in Lopez et al ref 10).
For clarity, I rephrased the sentence in conclusion to < The key element for predicting septic shock and in-hospital mortality is persistent infection at 7 days, and not persistent bacteremia at 48-72 hours>.
- Conclusions line 333-334: A formal recommendation for repeating blood cultures is not present in current guidelines (ESC 2023) therefore this sentence could be rephrased or removed.
Authors answer: in the American guidelines (2015) the formal class IIa indication was given in the text (see page 1445 in the Circulation paper), in the current ESC guideline 2023 I must admit that no formal recommendation was given, it is described under the key messages in chapter 15, that the duration of AB therapy is based on the first day of a negative blood culture.
I adapted the last sentence of conclusions: < Standard surveying blood cultures after starting antibiotic therapy in endocarditis patients should therefore not be advised >. I also adapted the abstract first sentence, <Background: previous guidelines for endocarditis have suggested repeating blood cultures until they become negative, with limited evidence>.
- Conclusions line 335-337: i believe this sentence should be rephrased because surgical indication are already “not only” related on persistent positive BC at day 3. Moreover, current guidelines (ESC 2023) state that the definition of persistent infection refers to positive BC at day 7 (as also stated by the authors in table 1).
Authors response: thank you, I agree, I adapted the sentence according to your point 4, and have deleted the sentence.
- Flowchart: authors could consider summarizing their clinical approach, based on the evidences they provided, in repeating BC in patients with IE using a flowchart underlying those cases in which repeating BC may be more relevant (S. aureus IE, ongoing sepsis despite pathogen-directed antimicrobial therapy, etc) vs those cases in which repeating BC may be avoided (Streptoccal IE, good clinical and biochemical response to antimicrobials, etc). However it is important to underline that those conclusions are made on expert opinion and should not be generalized to all IE cases.
Authors response: I have thought of this possibility, eg in MRSA and in sepsis. I think that the information provided in the text on persistent positive blood cultures in MRSA is warning readers that in this context, we may expect persistent cultures, but when or how to deal with the results is not known; in patients with sepsis it is quite normal to repeat blood cultures if only to look for other bacteria. So I did not think this was a good idea to give a non-evidenced based advice.
Minor comments: 1 . Line 201: Do the authors refer to high level aminoglycosides resistance (HLAR)?
Authors answer: Indeed, the resistance is described as partial for penicillins and cephalosporins, but there is also the recent emergence of high level aminoglycoside resistance (from reference 25). I adapted the sentence as: <Enterococcus faecalis is prevalent among the causative agents of endocarditis but is also known for partial antibiotic resistance against penicillin and ampicillin and recently emerging high level resistance to aminoglycosides.>
Reviewer 2 Report
Comments and Suggestions for Authors
The authors have put together a detailed reviewed literature on the incidence of persistent bacteremia and its association with outcomes, and the timing of valve culture negativization. The conclusion is that persistent infection at seven days is a more critical indicator than blood culture, and there is no need for ongoing blood culture surveillance after starting therapy.
I would recommend the review to be published after the authors address some minor comments:
1. Except for S. aureus, are there any reports showing the persistence of other Staphylococcus species and Streptococcus species in endocarditis patients?
2. Line 167- Write the year for Lopez et al.
3. Table 2- How was the percentage of native culture and patients with valve culture calculated? I would recommend including the number and in brackets the percentage
4. Please Italisize the name of the bacteria and the first letter of the genus name should be capitalized.
Comments on the Quality of English LanguageThe manuscript is well-written with clear and precise language. Only a few minor grammatical need to be addressed to improve clarity.
Author Response
Reviewer 2
Thank you for your advice and kind reading.
- Except for S. aureus, are there any reports showing the persistence of other Staphylococcus species and Streptococcus species in endocarditis patients?
Authors response: I could not find any reports on persistence of other Staphylococcus species. The persistence of Enterococcus faecalis bacteremia was however described (lines 204-214), and for Streptococcus bovis and Streptococcus viridans (lines 215-216) and for a group of others (lines 216-218).
- Line 167- Write the year for Lopez et al.
Authors response: I added the year 2013.
- Table 2- How was the percentage of native culture and patients with valve culture calculated? I would recommend including the number and in brackets the percentage
Authors response: the second column in table 2 refers to the total number of patients and the % of patients having native valves. The third column in table 2 refers to the % of patients having positive valve cultures. I adapted the third column by presenting the number of patients and added the % of the total number of patients in brackets.
- Please Italisize the name of the bacteria and the first letter of the genus name should be capitalized
Authors response: thank you for the correction. I changed it throughout the paper.
Round 2
Reviewer 1 Report
Comments and Suggestions for Authors
No further comments